# Prevalence of Burnout Syndrome in COVID-19 and Non-COVID-19 Units in University Hospital: A Cross-Sectional Study

**DOI:** 10.3390/ijerph191912664

**Published:** 2022-10-03

**Authors:** Romana Ulbrichtova, Viera Svihrova, Maria Tatarkova, Jan Svihra, Martin Novak, Henrieta Hudeckova

**Affiliations:** 1Department of Public Health, Jessenius Faculty of Medicine in Martin, Comenius University in Bratislava, Mala Hora 11149/4B, 036 01 Martin, Slovakia; 2Clinic of Urology, Jessenius Faculty of Medicine in Martin, Comenius University in Bratislava, Kollarova 2, 036 59 Martin, Slovakia

**Keywords:** COVID-19, burnout syndrome, healthcare workers, nurses

## Abstract

The aim of this study was to assess prevalence and associated risk factors of burnout syndrome among healthcare workers (HCWs), especially among nurses during the pandemic of COVID-19. The sample of the cross-sectional study consists of 201 employees of University Hospital. The Maslach Burnout Inventory—Human Services Survey for Medical Personnel (MBI–HSS MP) was used. An anonymous questionnaire was administered between 15 January and 1 February 2022. The majority of HCWs were female (79.4%). Overall, 69.2% displayed high levels of emotional exhaustion (EE), 35.3% high levels of depersonalization (DP), and 35.5% low levels of personal accomplishment (PA). Burnout was frequent among staff working in COVID units (EE 76.1%; DP 47.8%; and PA 46.7%). Burnout in EE and DP (70.7% and 36.6%, respectively) significantly prevailed in nurses working in COVID-19 units compared to non-frontline nurses (59.6 and 21.1%, respectively). Prevalence of burnout in PA was significantly higher in nurses working in non-COVID-19 units (47.4% vs. 29.3%). It is crucial to pay attention to the high prevalence of burnout syndrome in HCWs, especially in nurses, and not only in the frontline.

## 1. Introduction

The COVID-19 pandemic has been with us for more than two years and it has significantly affected our lives, and it has affected every section of the population in different ways.

Nurses are considered to be the most important members of any healthcare system, especially in the fight against COVID-19. In the system which provides healthcare in Slovakia, there is a long-term shortage of some healthcare workers (HCWs), mostly nurses. The lack of HCWs may threaten the accessibility of healthcare operations and their quality. Compared to countries of the Organisation for Economic Co-operation and Development (OECD), Slovakia lacks 13,874 nurses. For comparison, in Slovakia there are 5.7 nurses per thousand inhabitants, in Germany it is 13.9 nurses, in Austria 10.4, and in the neighbouring Czech Republic 8.6 and in Hungary 6.6 nurses. We consider that the number of missing nurses is currently significantly higher [1].

Work–life balance, high working pace, shifts, lack of sleep, limited resources, and occupational risk factors contribute to the adverse psychological consequences of being a HCW, in particular, insomnia, anxiety, and depression. According to the study conducted by Dall’Ora et al., patterns identified across 91 studies consistently show that adverse job characteristics (high workload, low staffing levels, long shifts, low control, low schedule flexibility, time pressure, high job and psychological demands, low task variety, role conflict, low autonomy, negative nurse–physician relationship, poor supervisor/leader support, poor leadership, negative team relationship, and job insecurity) are associated with burnout in nursing [2]. Nurses are generally considered to be a vulnerable group of the population in relation to physical and mental exhaustion, and this situation multiplied several times during the COVID-19 pandemic. Frontline nurses working in COVID-19 units are directly involved in the management and diagnostics of COVID-19. For this reason, they are more exposed to enormous stress due to ethical and moral dilemmas [3]. These employees need to decide matters of life and death quickly, and without protocols and guidelines. Other concerns are the experience of pain due to the loss of patients or colleagues and the high risk of infection and subsequent fear of contact with the families of infected patients. Burnout is included in the 11th Revision of the International Classification of Diseases (ICD-11) as an occupational phenomenon. It is not classified as a medical condition. Burnout is a syndrome conceptualized as resulting from chronic workplace stress that has not been successfully managed [4]. Burnout is a psychological syndrome arising from a long-term response to chronic interpersonal stressors at work that induce emotional exhaustion (EE), depersonalization (DP), and a sense of reduced personal accomplishment (PA) [5]. The burnout syndrome is a significant problem in modern workplace environments and its prevalence has increased substantially during the COVID-19 pandemic. We distinguished the following groups of symptoms of burnout syndrome: emotional problems (anxiety and depression), psychosomatic problems (weakness and insomnia), behavioural problems (aggression, irritability, and isolation), attitude problems (apathy and distrust), and other symptoms [6,7].

The aim of this study was to assess the prevalence and associated risk factors of burnout syndrome among HCWs at University Hospital, especially among nurses, during the pandemic of COVID-19.

## 2. Materials and Methods

### 2.1. Study Population

A cross-sectional study was conducted from 15 January to 1 February 2022 using an anonymous questionnaire distributed to individual email addresses and by QR codes. Employees were instructed not to complete the questionnaire more than once. Data were collected from all employees of University Hospital (*n* = 2318). Our sample size was estimated to be 92. This was calculated using a margin of error of 10 and based on the prevalence of burnout among HCWs before the COVID-19 pandemic [8]. The study was completed by 201 employees. The majority of HCWs were females (79.4%). The study protocol was approved by the Ethics Committee at the Comenius University in Bratislava, Jessenius Faculty of Medicine in Martin (reference no. EK 138/2018).

### 2.2. Measurement Tools

Burnout was assessed by using the Maslach Burnout Inventory—Human Services Survey for Medical Personnel (MBI—HSS MP). The questionnaire was developed by Christine Maslach and Susan E. Jackson to measure severity of mental burnout (1981), which later (1986, 1996) was modified for different professions—HCWs, teachers, social workers, and others. The burnout was measured according to the MBI—HSS [9,10]. The questionnaire was validated in Slovak and we obtained a license to use it for research purposes. The questionnaire was divided into three sections: (1) personal information including sex, education, age, duration of practice both in healthcare and in the university hospital, job, managerial job type, night-shift work, and the healthcare sector in the hospital; (2) MBI—Human Services Survey; (3) two closed-ended questions including history of COVID-19 and working in COVID-19 units.

The instrument used an eight-point Likert scale (intensity of feelings) starting from none (0) to maximal (7). The MBI—HSS comprised three subscales: emotional exhaustion (EE) with nine items (1, 2, 3, 6, 8, 13, 14, 16, and 20), depersonalization (DP) with five items (5, 10, 11, 15, and 22), and, lastly, the personal accomplishment (PA) dimension, encompassing eight items (4, 7, 9, 12, 17, 18, 19, and 21). The sum score of EE ranges from 0 to 63, in the subscale DP from 0 to 35, and in the PA from 0 to 54. The subscale score thresholds for identifying burnout were as follows: EE ≥ 27, DP ≥ 13, and PA ≤ 39 [9].

### 2.3. Internal Reliability

We calculated the internal reliability of the MBI—HSS using Cronbach’s coefficient alpha, which yielded the estimates 0.897 for EE, 0.799 for DP, and 0.826 for PA (Table 1).

### 2.4. Statistical Analysis

Statistical analyses were carried out using Epi Info 7 and SPSS 24. Descriptive statistics were given by frequencies and percentages. Comparisons between categorical variables were performed by chi-square or Fisher’s exact tests. The student’s *t*-test was used for comparison between two means for quantitative variables. The association between burnout scores and variables was explored by linear regression models. A p value of less than 0.05 was considered statistically significant.

## 3. Results

The demographic characteristics of the participants are shown in Table 2. A total of 201 respondents was considered for the analysis, of which 92 (45.8%) worked during the COVID-19 pandemic in the COVID unit. There were 158 (78.6%) females and 43 (21.4%) males. Among all respondents, 101 (50.3%) were physicians, 98 (48.8%) were nurses, and 2 (0.9%) were non-HCWs.

More than 69% of all respondents reported high levels of EE, more than a fifth reported high levels of DP, and more than 35% reported low levels of PA. We recorded more serious results in the subscale EE, DP, and PA among staff of the COVID-19 unit compared to the non-COVID-19 unit (Table 3a). There were statistically significant differences in the mean scores of EE, DP, and PA dimensions between nurses who worked in COVID-19 units and non-COVID-19 units (p = 0.009; 0.044; 0.001). Burnout in EE and DP (70.7% and 36.6%, respectively) prevailed in nurses working in COVID-19 units compared to non-frontline nurses (59.6 and 21.1%, respectively). Prevalence of burnout in PA was higher in nurses working in non-COVID-19 units (47.4% vs. 29.3%, respectively) (Table 3b).

The effect of variables on each score was examined using linear regression analysis among all respondents as presented in Table 4a and among nurses as presented in Table 4b.

## 4. Discussion

The COVID-19 pandemic hit us all over the world with tremendous force. Similar pandemics have occurred in the past. Research from the period of severe acute respiratory syndrome (SARS) or Middle East respiratory syndrome (MERS) outbreaks confirms the increased demands (burnout syndrome) on the frontline HCWs [11].

Based on MBI results, we can consider the COVID-19 pandemic as an emotional and physical stressful event. Slovakia is one of the most severely affected countries in terms of hospital overload and pressure on HCWs. Currently, we have recorded a total of 19,500 deaths from COVID-19 in Slovakia [12].

A large number of studies are currently under way to address burnout in HCWs during the COVID-19 pandemic. The results of several studies indicate high burnout rates among HCWs involved in the care of patients with COVID-19 [13,14,15,16]. Our results revealed that almost two thirds of the respondents recorded high EE, more than one third of the respondents recorded high DP and low PA, respectively, and were consistent with other studies in Italy, Egypt, and Germany [15,17,18]. The level of burnout syndrome also varies from the perspective of the type of profession. According to a systematic review of 52 articles, 38 studies examined burnout among psychotherapists [19]. The results of a Greek study among dentists indicate that physical and emotional exhaustion were at a very high level and were 5.5 and 8.5 times up, respectively, during the pandemic compared with before [20]. In the present work, nurses were more prone to burnout compared to physicians. Frontline nurses recorded higher average scores and high scores of EE and DP were more frequent among them.

The results of the burnout level among HCWs in Slovakia before the pandemic are presented by the authors Morovicsova et al. [10]. In our cohort, we found significant differences in frontline HCWs, suggesting that the growing burnout syndrome among HCWs is clearly associated with the COVID-19 pandemic. These results are also confirmed by Barello et al. (2020) where levels of EE appeared higher than normative values compared with the findings in other Italian samples before the COVID-19 outbreak [13,21]. Conversely, in some studies, the authors do not report an increased risk of burnout in connection with the COVID-19 pandemic [22,23]. According to an umbrella review of systematic reviews and meta-analyses among medical nurses, the prevalence of emotional exhaustion varied from 28–31%, the prevalence of depersonalization varied from 15–24%, and the prevalence of low personal accomplishment varied from 70–25% before the COVID-19 pandemic [24]. The findings of other meta-analyses with 45,539 nurses worldwide in 49 countries suggest that nurses have a prevalence of high burnout symptoms warranting attention and amelioration. An overall pooled-prevalence before the pandemic of COVID-19 of burnout symptoms among global nurses was 11.23% [25].

Our findings, that burnout is more prevalent among frontline HCWs, especially among nurses, compared with staff working in non-COVID units confirmed authors Lasalvia et al. [26]. This result contrasts with a study in China, reporting that frontline HCWs had a lower frequency of burnout compared with staff in a non-COVID unit [27]. However, even in our group, prevalence of burnout in PA was higher in nurses working in non-COVID-19 units compared to frontline nurses (47.4% vs. 29.3%, respectively).

The examined hospital was the frontline treatment center for COVID-19, but it was still open and some departments still operated in normal mode for a significant catchment area. Unfortunately, a multitude of organizational changes had had an impact on the hospital, including the displacement of nurses to other departments and the reduction of medical procedures due to the interruption of outpatient activities. Conversely, it is necessary not only to pay attention to the nurses working in the COVID-19 units, but also to the nurses who remained working in the non-COVID-19 units, where we likewise see a high level of workload, based on our results. The key reasons are the lack of HCWs, especially nurses in Slovakia, but also their allocation to COVID-19 units.

It is these facts that contribute to the frustrations of nurses during the pandemic and to increase of burnout. In this sense, the concept paper conducted by Parola et al. is evidence of the need for emotional support for healthcare team members [28]. Consequently, there is the need for designing action plans for burnout prevention and creating a healthy environment in hospitals.

In Slovakia and eight other countries of the European Union, burnout syndrome can be acknowledged as an occupational disease. Uniquely in Latvia, burnout syndrome is explicitly listed on the List of Occupational Diseases. Slovakia accepts chronic stress-related occupational diseases as an occupational risk through the “open item” on the List of Occupational Diseases [29,30].

The main limitation of our study is the subjectivity of the data obtained from the respondents. It was a cross-sectional study. It examined a specific population. The study was based only on a questionnaire evaluation without an objective evaluation. Because we conducted the research during the ongoing strong wave of the pandemic, the response rate was relatively low (8.7%). The reasons for this were the significant absence of medical personnel at work due to incapacity for work, quarantine, and, also, the enormous overload of medical personnel. In Slovakia, there is only a small amount of research devoted to the issue of burnout syndrome among HCWs. For this reason, we consider our results to be unique.

## 5. Conclusions

Even in this case, it has been clearly confirmed that pandemics lead to victims not only among patients, but also among HCWs. We recorded more serious results in all three subscales among HCWs in the COVID-19 unit compared to HCWs in the non-COVID-19 unit. Our findings clearly confirmed the high incidence of burnout in EE and DP dimensions among frontline nurses working in COVID-19 units during the COVID-19 outbreak. Non-frontline nurses showed worse mental-health outcomes in terms of higher PA. The impact of the COVID-19 pandemic is more emotionally burdensome for managerial frontline nurses, and, significantly, especially for those working on night shifts. Based on these findings, attention should be paid to addressing the high prevalence of burnout among HCWs, not only in the frontline and during pandemics.

## Figures and Tables

**Table 1 ijerph-19-12664-t001:** Cronbach’s alpha value of MBI and its subscales.

Subscales	Total	COVID Department	Non-COVID Department
**EE**	0.897	0.883	0.899
**DP**	0.799	0.807	0.773
**PA**	0.826	0.816	0.833

Note: EE, emotional exhaustion; DP, depersonalization; and PA, personal accomplishment.

**Table 2 ijerph-19-12664-t002:** Demographic characteristics of study population (*n* = 201).

Variable	All Respondents (*n* = 201)	COVID Unit (*n* = 92)	Non-COVID Unit (*n* = 109)	*p*-Value
**Working experience** **(years; average ± SD)**	20.05 ± 13.04	20.03 ± 13.07	20.0 ± 13.05	**0.036 ***
**Age (years)**				
**<35 *n* (%)**	57	30 (52.6)	27 (47.4)	0.219
**35–45 *n* (%)**	56	26 (46.4)	30 (53.6)	0.907
**>45 *n* (%)**	88	36 (40.9)	52 (59.1)	0.220
**Males *n* (%)**	43	15 (34.9)	28 (65.1)	0.106
**Females *n* (%)**	158	77 (43.7)	81 (56.3)	
**Physicians *n* (%)**	101	51 (50.5)	50 (49.5)	0.176
**N** **urses *n* (%)**	98	41 (41.8)	57 (58.2)	0.274
**N** **on-HCWs *n* (%)**	2	0 (0)	2 (100)	0.501
**Inpatient sector *n* (%)**	89	49 (55.1)	40 (44.9)	**0.018 ****
**Outpatient sector *n* (%)**	20	5 (25)	15 (75)	**0.049 ****
**Combined sector *n* (%)**	92	38 (41.3)	54 (58.7)	0.242
**Managerial position *n* (%)**	46	14 (34.4)	32 (65.6)	**0.017 ****
**N** **on-managerial position *n* (%)**	155	78 (50.3)	77 (49.7)	
**History of COVID *n* (%)**	90	50 (55.6)	40 (44.4)	**0.012 ****
**No history of COVID *n* (%)**	111	42 (37.8)	69 (62.2)	
**Night work *n* (%)**	136	74 (54.4)	62 (45.6)	**<0.001 ****
**No night work *n* (%)**	65	18 (27.7)	47 (72.3)	
**University degree *n* (%)**	158	74 (46.8)	84 (53.2)	0.562
**High school degree *n* (%)**	43	18 (41.9)	25 (58.1)	

Note: * *p* < 0.05 (student’s *t*-test); ** *p* < 0.05 (chi-square test).

**Table 3 ijerph-19-12664-t003:** (**a**) Levels of burnout syndrome according to the results of MBI-HSS: COVID and non-COVID unit. (**b**) Levels of burnout syndrome according to the results of MBI-HSS: COVID and non-COVID unit among nurses.

**(a)**				
	**All Respondents** **(*n* = 201)**	**COVID Unit** **(*n* = 92)**	**Non-COVID Unit** **(*n* = 109)**	***p*-Value**
**EE***n* (%)				
Low	25 (12.4)	6 (6.5)	19 (17.4)	**0.019 ***
Moderate	37 (18.4)	16 (17.4)	21 (19.3)	0.732
High	139 (69.2)	70 (76.1)	69 (63.3)	**<0.001 ***
**Mean score** (average ± SD)	33.1 ± 13.1	36.4 ± 12.5	30.3 ± 12.8	**<0.001 ***
**DP***n* (%)				
Low	81 (40.3)	31 (33.7)	50 (45.9)	0.079
Moderate	49 (24.4)	17 (18.5)	32 (29.4)	0.073
High	71 (35.3)	44 (47.8)	27 (24.7)	**<0.001 ***
**Mean score** (average ± SD)	9.9 ± 7.8	12.01 ± 8.1	8.2 ± 7.1	**<0.001 ***
**PA***n* (%)				
Low	71 (35.3)	43 (46.7)	28 (25.7)	**0.002 ***
Moderate	53 (26.4)	21 (22.8)	32 (29.4)	0.295
High	77 (38.3)	28 (30.4)	49 (44.9)	**0.034 ***
**Mean score** (average ± SD)	35.1 ± 8.8	33.8 ± 8.9	36.3 ± 8.6	**0.022 ***
**(b)**				
	**Nurses** **(*n* = 98)**	**COVID Unit** **(*n* = 41)**	**Non-COVID Unit** **(*n* = 57)**	***p*-Value**
**EE***n* (%)				
Low	16 (16.3)	4 (9.8)	11 (19.3)	0.196
Moderate	19 (19.4)	8 (19.5)	12 (21.1)	0.852
High	63 (64.3)	29 (70.7)	34 (59.6)	0.258
**Mean score** (average ± SD)	31.6 ± 13.1	35.2 ± 13.8	28.9 ± 11.9	**0.009 ***
**DP***n* (%)				
Low	51 (52.0)	21 (51.2)	30 (52.6)	0.890
Moderate	20 (20.4)	5 (12.2)	15 (26.3)	0.087
High	27 (26.7)	15 (36.6)	12 (21.1)	0.089
**Mean score** (average ± SD)	8.2 ± 7.3	9.7 ± 7.9	7.1 ± 6.7	**0.044 ***
**PA***n* (%)				
Low	35 (35.7)	22 (53.6)	13 (22.8)	**0.001 ***
Moderate	24 (24.5)	7 (17.1)	17 (29.8)	0.147
High	39 (39.8)	12 (29.3)	27 (47.4)	0.071
**Mean score** (average ± SD)	35.6 ± 8.1	33.6 ± 8.7	37.1 ± 7.3	**0.018 ***

Note: EE, emotional exhaustion; DP, depersonalization; PA, personal accomplishment; and * *p* < 0.05.

**Table 4 ijerph-19-12664-t004:** (**a**) Examination of the effect of variables on MBI scores (using linear regression analysis) among all respondents. (**b**) Examination of the effect of variables on MBI scores (using linear regression analysis) among nurses.

**(a)**				
**Variable**	**Coefficient**	**95% CI**	**Limits**	***p*-Value**
**Score EE**				
Gender (Female = 1; Male = 0)	3.017	−1.559	7.592	0.195
Position (Managerial = 1, Non-managerial = 0)	0.049	−4.263	4.361	0.982
Professions (Nurses = 1; Physicians = 0)	−3.492	−7.273	0.288	**0.070 ***
History of COVID-19 (Yes = 1; No = 0)	0.285	−3.411	3.980	0.879
COVID unit (Yes = 1; No = 0)	5.421	1.682	9.161	**0.005 ***
Correlation Coefficient r^2^	**0.07**			
**Score DP**				
Gender (Female = 1; Male = 0)	−1.506	−4.184	1.173	0.269
Position (Managerial = 1, Non-managerial = 0)	0.530	−1.994	3.054	0.679
Professions (Nurses = 1; Physicians = 0)	−2.940	−5.153	−0.727	**0.009 ***
History of COVID-19 (Yes = 1; No = 0)	0.039	−2.124	2.203	0.971
COVID unit (Yes = 1; No = 0)	3.765	1.575	5.954	**0.001 ***
Correlation Coefficient r^2^	**0.11**			
**Score PA**				
Gender (Female = 1; Male = 0)	−2.505	−5.656	0.646	0.118
Position (Managerial = 1, Non-managerial = 0)	2.219	−0.751	5.189	0.142
Professions (Nurses = 1; Physicians = 0)	1.140	−1.464	3.743	0.389
History of COVID-19 (Yes = 1; No = 0)	−0.501	−3.046	2.044	0.698
COVID unit (Yes = 1; No = 0)	−1.765	−4.340	0.811	0.178
Correlation Coefficient r^2^	**0.05**			
**(b)**				
**Variable**	**Coefficient**	**95%CI**	**Limits**	***p*-Value**
**Score EE**				
Age	−0.377	−1.112	0.358	0.310
Working experience	0.327	−0.318	0.972	0.316
Position (Managerial = 1, Non-managerial = 0)	6.506	−0.549	13.561	**0.070 ***
History of COVID-19 (Yes = 1; No = 0)	0.715	−4.704	6.133	0.793
COVID unit (Yes = 1; No = 0)	5.227	−0.444	10.898	**0.070 ***
Night shifts	6.258	0.264	12.253	**0.040 ***
Correlation Coefficient r^2^	**0.12**			
**Score DP**				
Age	0.196	−0.273	0.665	0.408
Working experience	−0.180	−0.591	0.231	0.386
Position (Managerial = 1, Non-managerial = 0)	−1.379	−5.876	3.118	0.544
History of COVID-19 (Yes = 1; No = 0)	0.315	−3.139	3.770	0.856
COVID unit (Yes = 1; No = 0)	−3.241	−6.855	0.374	**0.078 ***
Night shifts (Yes = 1; No = 0)	−1.986	−5.807	1.835	0.305
Correlation Coefficient r^2^	**0.06**			
**Score PA**				
Age	−0.078	−0.506	0.350	0.718
Working experience	0.010	−0.366	0.386	0.957
Position (Managerial = 1, Non-managerial = 0)	1.858	−2.250	2.068	0.371
History of COVID-19 (Yes = 1; No = 0)	−1.054	−4.209	1.588	0.508
COVID unit (Yes = 1; No = 0)	2.883	−0.419	1.622	**0.086 ***
Night shifts (Yes = 1; No = 0)	0.689	−2.801	1.757	0.695
Correlation Coefficient r^2^	**0.05**			

Note: EE, emotional exhaustion; DP, depersonalization; PA, personal accomplishment; HCWs, healthcare workers; and * *p* < 0.05.

## Data Availability

All data are fully available without any restriction upon reasonable request.

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
