# Peer review of "Prevalence of Burnout Syndrome in COVID-19 and Non-COVID-19 Units in University Hospital: A Cross-Sectional Study"

_ijerph, 2022, doi:10.3390/ijerph191912664_

Round 1
Reviewer 1 Report
Please see attached file

Author Response
Response to Reviewer 1 Comments
Dear reviewer,
thank you for your comments, which were very inspiring for us. We have edited the text of the manuscript according to your recommendations.
Point 1: “The good data analysis is misplaced due to these two "poor" parts. Also references need to be addressed refreshed since there are a limited…..”
Response 1: We added to Introduction:
According to the study conducted by Dall´Ora et al., patterns identified across 91 studies consistently show that adverse job characteristics (high workload, low staffing levels, long shifts, low control, low schedule flexibility, time pressure, high job and psychological demands, low task variety, role conflict, low autonomy, negative nurse-physician relationship, poor supervisor/leader support, poor leadership, negative team relationship, and job insecurity) are associated with burnout in nursing [2].
We added to Discussion:
According to systematic review of 52 articles, thirty-eight studies examined burnout among psychotherapists [19]. The results of Greek study among dentists indicate that physical and emotional exhaustion were at a very high level and were 5.5 and 8.5 times up respectively during the pandemic than before [20].
The findings of other meta-analyses with 45,539 nurses worldwide in 49 countries suggest that nurses have high burnout symptoms prevalence warranting attention and imple-mentation. An overall pooled-prevalence before the pandemic of COVID-19 of burnout symptoms among global nurses was 11.23% [25].
In this sense, the concept paper conducted by Parola et al., evidences the need for emo-tional support for healthcare team members [28].
In Slovakia and other 8 countries of European Union burnout syndrome can be acknowledged as an occupational disease. Uniquely in Latvia, burnout syndrome is explicitly listed on the List of Occupational Diseases. Slovakia accept chronic stress-related occupational diseases as occupational through the “open item” in the List of Occupational Diseases [29, 30].
Point 2: LINES 141-147 are a repetition of the results part. Try to discuss more the data by using more obvious to the reader way such as..”one out of three nurses were…..”
Response 2: We discussed our results: Our results revealed that almost two thirds of the respondents recorded high EE, more than one third of the respondents recorded high DP and low PA, respectively and were consistent with other studies in Italy, Egypt, and Germany [15, 17, 18].
Point 3: LINES 154-155.. Ι do not understand. Please verify what you mean
Response 3: We have reformulated the sentence: Conversely, in some studies, the authors do not report an increased risk of burnout in connection with the COVID-19 pandemic [22, 23].
Point 4: LINES 166-173 should go in the beginning of the discussion part. In any case the discussion needs rewritten according to
comments above.
Response 4: We replaced the mentioned parts at the beginning of the discussion.
Point 5: Spelling and English corrections needed
Response 5: We improved and corrected the English in the manuscript. The manuscript has been reviewed by an English speaker.
Point 6: Address whether there are only nurses or also other health professionals. In the abstract you mention “healthcare workers”,
elsewhere you say only nurses. There is a huge difference.
-If there are only nurses you should change the title of the article otherwise is misleading
Response 6: The majority of HCWs were physicians (n = 101), nurses (N = 98), and other respondents (N = 2). Nurses in general (also based on the available literature and also in our results) achieve worse results and a higher prevalence of burnout. For this reason, we focused on nurses and selected the subgroup of nurses.
We believe that this publication will bring new insight into the issue and will take appropriate action in the professional community.
Yours sincerely
Dr. Martin Novak

Reviewer 2 Report
1. This paper “Prevalence of burnout syndrome in COVID-19 and non 2 COVID-19 units in university hospital: a cross-sectional study” is well written, however, minor issues need to be fixed before publication process. My comments and suggestions are following:
2. Abstract does not show any novel hypothesis, question, or method? For a new research gaps, what is new?
3. What are practical policy implications?
4. Few formatting issues in the paper. Tables are scattered and also of different styles. Fix them during revision please.
5. Literature review is less, add more relevant literature and extend your introduction part.
6. Not even a single figure added. This makes presentation imbalanced.
7. Few typos exist in this paper, for instance, page 8, first paragraph.
8. Clear problem statement and rationale should be improved.
9. Look for redundancy of the references if any.
Author Response
Response to Reviewer 2 Comments
Dear reviewer,
thank you for your comments, which were very inspiring for us. We have edited the text of the manuscript according to your recommendations.
Point 1: Abstract does not show any novel hypothesis, question, or method? For a new research gaps, what is new?
Response 1: Thank you for your comment. The COVID-19 pandemic has been with us for more than two years and it has significantly affected our lives and it affected all part of the population in a different way. In the system of providing health care in Slovakia, there is a long-term lack of some healthcare workers (HCWs), mostly nurses. In Slovakia, there is only a small amount of research devoted to the issue of burnout syndrome among health professionals. For this reason, we consider our results to be unique.
Point 2: What are practical policy implications?
Response 2: At the beginning of the introduction, we addressed this issue:
Nurses are considered to be the most important members of any healthcare system, especially in the fight against COVID-19. In the system of providing health care in Slovakia, there is a long-term lack of some healthcare workers (HCWs), mostly nurses. The lack of HCWs may threaten the accessibility of health care running and its quality. Compared to Organisation for Economic Co-operation and Development (OECD) countries, Slovakia lacks 13,874 nurses. For comparison, in Slovakia there are 5.7 nurses per thou-sand inhabitants, in Germany it is 13.9 nurses, in Austria 10.4, in the neighbouring Czech Republic 8.6 and in Hungary 6.6 nurses. We consider that the number of missing nurses is currently significant higher [1].
Point 3: Few formatting issues in the paper. Tables are scattered and also of different styles. Fix them during revision please.
Response 3: We have corrected the formatting of the text and tables.
Point 4: Literature review is less, add more relevant literature and extend your introduction part.
Response 4: We added to Introduction:
According to the study conducted by Dall´Ora et al., patterns identified across 91 studies consistently show that adverse job characteristics (high workload, low staffing levels, long shifts, low control, low schedule flexibility, time pressure, high job and psychological demands, low task variety, role conflict, low autonomy, negative nurse-physician relationship, poor supervisor/leader support, poor leadership, negative team relationship, and job insecurity) are associated with burnout in nursing [2].
We added to Discussion:
According to systematic review of 52 articles, thirty-eight studies examined burnout among psychotherapists [19]. The results of Greek study among dentists indicate that physical and emotional exhaustion were at a very high level and were 5.5 and 8.5 times up respectively during the pandemic than before [20].
The findings of other meta-analyses with 45,539 nurses worldwide in 49 countries suggest that nurses have high burnout symptoms prevalence warranting attention and implementation. An overall pooled-prevalence before the pandemic of COVID-19 of burnout symptoms among global nurses was 11.23% [25].
In this sense, the concept paper conducted by Parola et al., evidences the need for emotional support for healthcare team members [28].
In Slovakia and other 8 countries of European Union burnout syndrome can be acknowledged as an occupational disease. Uniquely in Latvia, burnout syndrome is explicitly listed on the List of Occupational Diseases. Slovakia accept chronic stress-related occupational diseases as occupational through the “open item” in the List of Occupational Diseases [29, 30].
Point 5: Not even a single figure added. This makes presentation imbalanced.
Response 5: Thank for your comment. We have used tables to present the results for better clarity with respect to the data that is presented.
Point 6: Few typos exist in this paper, for instance, page 8, first paragraph.
Response 6: We have corrected the errors and typos in the text. We improved and corrected the English in the manuscript. The manuscript has been reviewed by an English speaker.
Point 7: Clear problem statement and rationale should be improved.
Response 7: This section has been edited.
Point 8: Look for redundancy of the references if any.
Response 8: Thank you for the comment, we have edited and updated the literature according to the reviewer's comments and in accordance with the instructions for authors.
We believe that this publication will bring new insight into the issue and will take appropriate action in the professional community.
Yours sincerely
Dr. Martin Novak

Reviewer 3 Report
Your paper is very interesting but there are some corrections-suggestions to make
1) In the abstract we write in the present tense
2) in line 14 is better to include the word "sample"
3) line 22 and 23 could be written in a better way avoiding the word "should"
4)line 27-28 be careful to use the same tense, verbs are presented in different tenses
5) line 30-31, It is difficult to understand the meaning. Please rewrite the sentence
6) there is an enormous amount of grammar and syntactic structure that in many cases affect the meaning of the text. The whole text needs to be corrected
Author Response
Response to Reviewer 3 Comments
Dear reviewer,
thank you for your comments, which were very inspiring for us. We have edited the text of the manuscript according to your recommendations.
Point 1: In the abstract we write in the present tense
Response 1: Thank you for the comment. Based on the abstracts of the articles that are published in the IJERPH, we used the past tense or indefinite tense.
Point 2: in line 14 is better to include the word "sample"
Response 2: We used “sample”: The sample of the cross-sectional study consists of 201 employees of the University Hospital.
Point 3: line 22 and 23 could be written in a better way avoiding the word "should"
Response 3: We changed this sentence: It is crucial to pay attention to the high prevalence of burnout syndrome in nurses, not only in the front-line.
Point 4: line 27-28 be careful to use the same tense, verbs are presented in different tenses
Response 4: We corrected this sentence: The COVID-19 pandemic has been with us for more than two years and it has significantly affected our lives and it affected all part of the population in a different way.
Point 5: line 30-31, It is difficult to understand the meaning. Please rewrite the sentence
Response 5: We rewrote the sentence: In the system of providing health care in Slovakia, there is a long-term lack of some healthcare workers (HCWs), mostly nurses. The lack of HCWs may threaten the accessibility of health care running and its quality.
Point 6: there is an enormous amount of grammar and syntactic structure that in many cases affect the meaning of the text. The whole text needs to be corrected
Response 6: We improved and corrected the English in the manuscript. The manuscript has been reviewed by an English speaker.
We believe that this publication will bring new insight into the issue and will take appropriate action in the professional community.
Yours sincerely
Dr. Martin Novak

Reviewer 4 Report
1. Was the questionnaire validated in Slovak?
2. Results. Line 108 (0.3%) - incorrect percentage value.
3. In Table 2, the p-value for the 'physicians' variable (p = 1.825) is incorrect.
4. Please explain any abbreviations in the tables in the footnotes.
5. Why did the authors only select the subgroup of nurses in Tables 3 and 4?
6. What is the basis of the variable selection for the regression model?
7. What was the workload in Covid units?
8. Did the speciality of the non-COVID unit matter?
9. What was the care of COVID patients in the outpatient sector?
10. What was the workload of the participants (how many patients were per person, the severity of the COVID course, number of deaths, etc.)?
11. Table 3a. Please check the numbers in the tables again, there are errors in the numbers, eg EE-> High (COVID unit) should be the number 70, not 7.
12. In the discussion, please consider the questionnaire's components (EE, PA and DP) and the reasons for the observed results.
13. What are the cut-off points for the diagnosis of burnout for the Slovak population (low, moderate, high risk)?
Author Response
Response to Reviewer 4 Comments
Dear reviewer,
thank you for your comments, which were very inspiring for us. We have edited the text of the manuscript according to your recommendations.
Point 1: Was the questionnaire validated in Slovak?
Response 1: Yes, the questionnaire is validated in Slovak. (https://www.mindgarden.com/314-mbi-human-services-survey). We have purchased a license to use the questionnaire.
We added this information in Methods: The questionnaire is validated in Slovak and we have obtained a license to use it for research purposes.
Point 2: Results. Line 108 (0.3%) - incorrect percentage value.
Response 2: We corrected the percentage value: Among all respondents, 101 (50.3%)…….
Point 3: In Table 2, the p-value for the 'physicians' variable (p = 1.825) is incorrect.
Response 3: We corrected the p-value: p = 0.176.
Point 4: Please explain any abbreviations in the tables in the footnotes.
Response 4: All abbreviations are explained below the table in the footnotes.
Point 5: Why did the authors only select the subgroup of nurses in Tables 3 and 4?
Response 5: Nurses in general (also based on the available literature and also based on our results) achieve worse results and a higher prevalence of burnout. For this reason, we focused on nurses and selected the subgroup of nurses.
Point 6: What is the basis of the variable selection for the regression model?
Response 6: We selected variables that are already proven in the literature to be related to the outcome and variables that can either be considered the cause of the exposure, the outcome, or both.
Point 7: What was the workload in Covid units?
Response 7: At the beginning of the introduction, we addressed this issue:
Work-life balance, high working pace, shifts, lack of sleep, limited resources and occupational risk factors contribute to the adverse psychological consequences between HCWs, in particular insomnia, anxiety and depression. According to the study conducted by Dall´Ora et al., patterns identified across 91 studies consistently show that adverse job characteristics (high workload, low staffing levels, long shifts, low control, low schedule flexibility, time pressure, high job and psychological demands, low task variety, role conflict, low autonomy, negative nurse-physician relationship, poor supervisor/leader sup-port, poor leadership, negative team relationship, and job insecurity) are associated with burnout in nursing [2].
Point 8: Did the speciality of the non-COVID unit matter?
Response 8: We considered units/wards where "white" medicine was performed and workplaces were not reprofiled as a non-COVID unit.
Point 9: What was the care of COVID patients in the outpatient sector?
Response 9: Thanks for your comments, the aim of the study was to assess prevalence and associate risk factors of burnout syndrome among HCWs, especially among nurses in University hospital in Martin-inpatient clinic.
The impact of the COVID-19 pandemic on healthcare utilization, especially for outpatient care was significant. The COVID-19 pandemic has dramatically changed how outpatient care is delivered in health care practices. To decrease the risk of transmitting the virus to either patients or health care workers within their practice, providers were deferring elective and preventive visits. Many patients were also avoiding visits because they did not want to leave their homes and risked exposure. Also influencing both outpatient healthcare worker and patient behaviour were the evolving recommendations and restrictions.
Point 10: What was the workload of the participants (how many patients were per person, the severity of the COVID course, number of deaths, etc.)?
Response 10: Thank you for your comment. We did not address this issue, but for your information we found the following data.
The COVID-19 pandemic had a huge impact on the healthcare industry during 2021. Due to the rush of patients with COVID-19 requiring hospitalization, the Ministry of Health of the Slovak Republic had to proceed with the reprofiling of several departments of hospitals. Number of hospitalized cases of patients with COVID-19 during 2021 in University Hospital in Martin: 1,303 (according to the Health care surveillance authority).
Point 11: Table 3a. Please check the numbers in the tables again, there are errors in the numbers, eg EE-> High (COVID unit) should be the number 70, not 7.
Response 11: We checked the number in the tables.
Point 12: In the discussion, please consider the questionnaire's components (EE, PA and DP) and the reasons for the observed results.
Response 12: MBI is characterized by three dimensions: EE, PA, and DP; we compared our results with the available results of other studies.
Point 13: What are the cut-off points for the diagnosis of burnout for the Slovak population (low, moderate, high risk)?
Response 13: Burnout is now categorized as a “syndrome” that results from “chronic workplace stress that has not been successfully managed,” according to the World Health Organization's International Disease Classification (ICD-11)—the official compendium of diseases.
Burnout syndrome is not currently often recognized as an occupational disease in Slovakia, but based on our results, it is necessary to solve this problem. Only in 39% of the countries a possibility to acknowledge burnout syndrome as an occupational disease exists, with most of compensated cases only occurring in recent years. New systems to collect data on suspected cases have been developed reflecting the growing recognition of the impact of the psychosocial work environment. In agreement with the EU legislation, all EU countries in the study have an action plan to prevent stress at the workplace.
We added to discussion: In Slovakia and other 8 countries of European Union burnout syndrome can be acknowledged as an occupational disease. Uniquely in Latvia, burnout syndrome is explicitly listed on the List of Occupational Diseases. Slovakia accept chronic stress-related occupational diseases as occupational through the “open item” in the List of Occupational Diseases [29, 30].
We believe that this publication will bring new insight into the issue and will take appropriate action in the professional community.
Yours sincerely
Dr. Martin Novak

Round 2
Reviewer 1 Report
No further changes. Thank you for your effort
Reviewer 3 Report
Yes I read the letter of the authors and the manuscript and it is improved so it can be published in the current form